# Development of Food Packaging through TRIZ and the Possibility of Open Innovation

Harry Jeong [1] , Seunggu Lee [2] and Kwangsoo Shin [1,*]

1   Department of Biomedical Convergence, College of Medicine, Chungbuk National University, Chungdae-ro 1, Seowon-gu, Cheongju 28644, Chungbuk, Korea; harry@g.cbnu.ac.kr
2   Dr. Chung's Food Co., Ltd., Sandan-ro 124, Heungdeok-gu, Cheongju 28446, Chungbuk, Korea; sglee@vegemil.co.kr
*   Correspondence: sksidea@chungbuk.ac.kr

**Abstract:** As the aging population increases, the need for new product development (NPD) for elderly-friendly food packaging is also increasing. Through the use of consumer research, this study derived the following problems when elderly people use food packaging: "The contents easily overflow when holding the container", "It is hard to pour", "Remnant remains after pouring", and "It is hard to use a straw". To address these problems, this study applied the following principles of TRIZ: principle 1 (segmentation) and principle 22 (blessing in disguise). In order to materialize the improvement plan, this study developed an elderly-friendly pouch-based packaging from the perspective of universal design. This study shows that it is possible to use the TRIZ technique in the NPD of food packaging, and that it is possible to secure commercial competitiveness from the view of universal design. This study is expected to serve as a starting point for further study on the NPD of elderly-friendly food packaging.

**Keywords:** open innovation; elderly friendly; senior friendly; universal design; aging; aging consumer; TRIZ; silver food; problem solving; aged

## 1. Introduction

With the advancing development of industrial and medical technology, the elderly population is continuously increasing. The World Health Organization (WHO) classifies those over 65 years of age as being elderly [1]. A decline in the birth rate and an increase in life expectancy are increasing the proportion of the elderly. This trend, which started in advanced countries, is spreading to developing countries. In the case of Korea, the elderly population was 15.7% in 2020 but is expected to exceed 25% in 2030 and 39.8% of the total population in 2050 [2]. Caring for the elderly is becoming a social problem due to the aging rate and increase in life expectancy.

Elderly people have limited physical activity due to aging. The overall function of their body organs is reduced, and their immunity is weakened, making them vulnerable to diseases [3]. Their number and duration of hospitalizations and their need for medical treatment increases, and their medical expenses generally increase as well. Aging is a time when proper nutrition and lifestyle management are necessary to maintain good health. Elderly people psychologically prefer familiar things to new adventures [4]. They have a fear of changing their existing choices because they have established their own lifestyle through long experience.

As products for the elderly have been researched and developed, an elderly-friendly industry has emerged, but its speed of development is very slow [5]. Research is beginning in some areas, such as the residential environment of the elderly and household items. Existing products that have been used for people with disabilities or for those with ill health have changed their names and are now being used for the elderly. Even if an elderly person has not been officially diagnosed as having a disability or disease, there

are limited products that can be used by those with weak physical functions. At a time when the elderly population is increasing, studies on products for the elderly are becoming increasingly necessary.

Research on food packaging for the elderly is also insufficient [6]. Although the use of processed food by the elderly is increasing, food packaging that considers the elderly is limited. Food packaging should be convenient and should provide various information about what is in the food. However, food packaging is often producer-oriented [7]. Although there are many difficulties faced by the physically weak, including the elderly, due to food packaging, there are few studies on food packaging.

In general, packaging protects the product contained therein and provides a safe and healthy way to transport or store it [8]. In particular, the package plays a very important role in the purchase decision [9,10]. Food packaging, which is directly related to health, protects food from foreign materials, prevents spoilage, and makes eating easier. It retains the benefits of food processing after the manufacturing is complete, allowing food to travel safely from its origin to its point of consumption [11]. Therefore, food packaging is important for maintaining food quality and safety [12]. In addition, it is necessary to consider user convenience in the development of food packaging [13]. In food packaging for the elderly, fundamentally reduced physical abilities must be considered. Food consumption by elderly people is made difficult by the problems of unpacking [14] and reading labels [15] in particular. In terms of food use, it is necessary to determine the problems that occur during the actual eating process and to solve them.

Therefore, this study determined the problems of food packaging for the elderly through a consumer survey. In addition, this study found a solution to the problem through TRIZ for pouch-based food packaging that the elderly frequently use, and built a practical improvement plan from the view of universal design. Wan et al. [16] applied the TRIZ theory to a new energy vehicle service and formed an idea according to service derivative characteristics. Their research validated the solution by examining the practical problems in developing the new energy vehicle market. Feniser et al. [17] used TRIZ to obtain efficient processes, products, and sustainable processes for small- and medium-sized enterprises (SMEs). In the context of TRIZ research, this study will contribute to further catalyzing the development of the elderly-friendly food industry, while making food consumption more convenient by developing a pouch-based packaging design.

The remainder of this study is organized as follows: It begins with a literature review covering elderly food packaging, universal design, TRIZ, and pouch-based food packaging. The subsequent section presents the methodology applied in this study. Section four presents the results, while section five and the final section offer a discussion and conclusions.

## 2. Theory and Background

### 2.1. The Need for the Development of Elderly-Friendly Food Packaging

As the proportion of the elderly population increases, elderly people seek not only to increase their lifespan but also to live a healthy life. However, it is difficult for the aged to maintain their wellbeing, due to deterioration in their physical function and economic poverty. The elderly group with reduced economic activity also falls into a vicious cycle of economic poverty because of increased medical expenses from chronic diseases. In order to maintain the health of elderly people, whose proportion is increasing in an aging society, it is first necessary to support appropriate medical services and nutritional management. In that respect, a standardized platform on nutrition and lifestyle of the elderly developed by Park et al. [18] is meaningful. Professional management programs for the elderly and training of professional personnel are also needed [19].

One of the major characteristics of the demographic change is that the number of the elderly one- or two-person households is increasing. Elderly people living alone or within a couple often use processed foods, such as HMR, rather than direct food cooking, which requires a lot of raw material costs and cooking time. Moreover, elderly people

often use processed foods as health supplements. In an aging society, the health care of elderly people is not only an individual problem but also a problem for society as a whole. The government should minimize social costs by promoting the health of elderly people. Convenience of use should be provided to the elderly who mainly use processed foods to help maintain their health. Improvement of food packaging for the elderly consumers is essential to solve the inconvenience for elderly people of using processed food.

*2.2. Universal Design*

According to White et al. [20], universal design is defined as a type of design that can accommodate all people (children, the elderly, people with disabilities, etc.), irrespective of gender and age, and provides a safe, functional, and accessible environment for all people. This type of design can be summarized as 'design for all'. Universal design is not simply a concept for people with low physical abilities but has been expanded to accommodate the entire human life cycle. Story [21] presented the seven principles pursued by universal design.

There is presently a demand for an environment in which all users can live easily and conveniently regardless of their age, gender, or disabilities. In NPD, the application of universal design has become a new paradigm for developing products that are comfortable and easy to use for all users. There is a close correlation between universal design and elderly-friendly products. As part of overcoming the physical and sensory changes that occur during the aging process, the independence and autonomy of the elderly can be achieved if elderly-friendly products are designed to be simpler to use and more modern. In addition, by improving the product's functions in consideration of the convenience of the elderly, one of the most vulnerable groups to use the product, the product can be positioned as a universal product for everyone.

Ericsson et al. [22] conducted a qualitative text analysis of universal design policy in Spain. The two tasks identified for universal design policy in the analysis were to convey that universal design is a type of design for everyone and to find a way to avoid normal thinking, instead focusing on innovation. There is a need to change the way universal design is conceptualized and the fundamental ways in which people are classified.

Theories for design improvements include 'design thinking' and 'design for X'(DFX). Design thinking is generally defined as an analytical and creative process that engages people in experimentation, model creation and prototyping, feedback, and redesign opportunities [23]. Most importantly, the ways in which professional designers solve problems are valuable for producing innovative companies and changing societies [24]. However, according to Kimbell [24], design thinking has problems. First, this type of thinking relies on a distinction between thinking, knowing, and acting in the world. Kimbell [24] also argued that generalized design thinking ignores the historically established diversity of designer practices and institutions. Third, design thinking relies on design theory that privileges designers as the primary actors in design. DFX is a generic framework that can be easily extended or tuned to enable the rapid development of various DFX tools with consistent quality, and provides many formal but practical structures [25]. DFX techniques are widely used in holistic product development, but each technique focuses on only one aspect of the product, such as the product's manufacturing or cost [26]. Universal design refers to the design of products that are usable by as many people as possible with little or no extra cost [27]. Thus, the use of universal design principles along with the open innovation concept can increase the possible sustainability goals of information systems, which can be achieved by improving the success of the system and increasing user satisfaction [28]. Since the customer base is becoming increasingly disabled, elderly, and socially disadvantaged, companies can maximize the potential market by using universal design starting from the early stages of product development [29]. Specifically, this study determined that universal design is suitable for businesses targeting the elderly.

### 2.3. Application of TRIZ to Develop a Universal Design

There are various types of problem-solving techniques including brainstorming, SCAMPER, and ASIT. Brainstorming, popularized by Osborn [30], is a technique in which group members seek solutions to problems by freely presenting ideas. However, brainstorming groups often have fewer and poorer-quality ideas than the same number of individuals working alone [31]. This technique, therefore, has difficulties in generating innovative ideas step by step [32]. Advanced Systematic Inventive Thinking (ASIT) is a problem-solving technique derived from TRIZ and developed to be easily utilized in the field [33]. However, because ASIT solutions are sought using objects available in the problem space of the system [34], it could be difficult to pursue open innovation using this method.

TRIZ (Teoriya Reshniya Izobretatelskikh Zadatch) is a Russian abbreviation for the theory of inventive problem solving [35]. This theory was established by G.S. Altshuller who discovered the existence of common and predictable patterns and laws in the development of science and technology. The existence of principles means that we possess a theoretical framework that can specifically explain a given problem and help engineers find creative ideas to solve that problem [35]. TRIZ promotes using the thinking of researchers to derive creative solutions and is very useful in solving scientific and technological problems because it is a problem-solving methodology based on invention patents [36].

TRIZ studied important patents from around the world, set 39 engineering parameters with technically contradictory relationships, and created a 'contradiction matrix' listing these parameters horizontally and vertically [37]. Most technical problems contain at least one contradiction. The concept of contradiction here means that an attempt to improve one element undermines another. The core of TRIZ is the idea that finding and solving the contradictions that cause problems can lead to innovative solutions to real problems. The 40 inventive principles of TRIZ provide designers clear guidance for resolving technical contradictions.

### 2.4. Pouch-Based Food Packaging

Pouch packaging is made from laminated materials with barrier properties on three or four layers. Laminated film can use various types of film composition depending on the packaging type and method. Materials with moisture resistance, gas barrier properties, and light barrier properties are commonly used. In particular, liquid food packaging must be able to withstand the strength of most automatic filling technologies and be suitable for high-speed filling.

Pouch packaging includes retort pouches, standing pouches, and spout pouches. A retort pouch is used for packaging retort food with sterilization functions. Retort food is heat-treated at a temperature of 100 °C or higher for a certain period of time after filling the food in a laminated packaging with aluminum foil. This heat treatment process is performed by adding steam or hot water at a constant pressure (1.5~3.0 kg/cm$^2$) above the atmospheric pressure in a pressure container called a retort. Standing pouches can be erected with the bottom side open. A spout pouch is a type of packaging with a cap attached to it. This type of packaging has a bottle function that allows the user to reseal the cap once opened.

### 2.5. Open Innovation

Open innovation can be defined as 'the use of purposive inflows and outflows of knowledge to accelerate internal innovation, and expand the markets for the external use of innovation, respectively' [38,39]. To promote innovation within a given organization, it is possible to expand the innovation created within that organization to include innovation from outside of the organization by utilizing the inflow and outflow of resources, such as internal and external technologies and ideas. This technique can maximize performance [40] and expand market opportunities by absorbing external ideas [41]. This paradigm assumes that firms can, and should, use both external ideas and internal ideas, as well as internal

and external paths to market, as firms look to advance their technologies [38]. As innovation that creates new customer value is no longer possible, open innovation that utilizes all available resources has come to be recognized as an essential strategy for firms [42]. The target of open innovation has been expanded from technology to product development and business models [43]. In addition to product and business models, innovations that include customer experiences are being proposed for constructing a service-providing platform [44].

Collaborative networks can improve innovative performance by providing inventors with the two types of knowledge they need to innovate [45]. First, collaborative networks can provide new knowledge that inventors can integrate and reconfigure to achieve innovation [46]. Second, collaborative networks can provide deep knowledge and the capability to integrate and reconstruct new knowledge [47]. Integrated knowledge refers to the implicit ability and understanding of how to combine and reconstruct new knowledge, which helps inventors create valuable innovations [47,48]. In recent years, firms have increasingly relied on external information [49] and R&D collaborations [50] to develop new products, services, and processes. Open innovation is designed to facilitate the introduction of new products and services by firms through the acquisition of external knowledge, thus saving significant cost and time [38,49]. At the same time, firms have become more active in licensing out their internally developed technology and selling it to external agents [51].

### *2.6. Hypothesis*

With an aging population and an increase in the number of patients, various processes are being attempted in the field of NPD [52]. Thus, there is no reason not to try new methods, including TRIZ, to design food packaging for the elderly. To respond to increasing demands, TRIZ could prove an alternative method of problem solving. In addition, an approach based on universal design will provide an optimal perspective for developing new products necessary for the elderly. Therefore, the present study proposed the following hypothesis related to food packaging for the elderly:

**Hypothesis 1:** *Improved food packaging from a universal design perspective can be developed by solving the problem of elderly-friendly food packaging by applying TRIZ.*

### 3. Methodology

### *3.1. Analytic Procezss*

In the first part of the study, a survey was conducted on elderly-friendly foods. During this step, we investigated the general consumer perception for elderly-friendly food and problems in use and then determined points to be improved. Next, we applied TRIZ to the technical contradictions that emerged from the questionnaire. Through this process, we developed an idea to solve the problem. Finally, elderly-friendly food packaging was developed by applying universal design, as summarized in Figure 1.

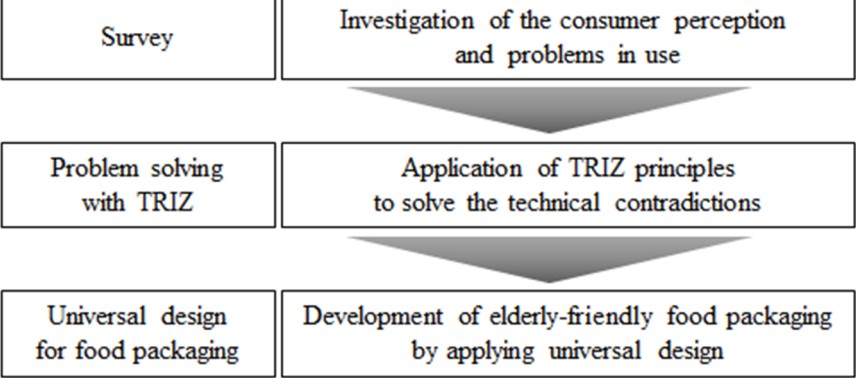

**Figure 1.** Research process.

*3.2. Variables*

To measure the research variables used in this study, we modified and applied the measurement items whose reliability and validity were already been verified in previous studies. Table 1 summarizes the measurement items based on the relevant literature for each factor.

**Table 1.** Measurement items of variables.

| Factor | | Measurement Items | References |
|---|---|---|---|
| Awareness of elderly-friendly food packaging | 1.<br>2.<br>3.<br>4.<br>5. | Awareness and method of awareness<br>Purchasing experience<br>Purchasing purpose<br>Main issues with elderly-friendly food<br>Need for elderly-friendly food packaging | [53] |
| Awareness of beverage packaging | 1.<br>2.<br>3. | Preferred beverage packaging<br>Preferred reason for and method of drinking a beverage<br>Items to be improved first in elderly-friendly food packaging | [54–57] |
| Claim for elderly-friendly pouch packaging | 1.<br>2.<br>3. | Disadvantages of pouch packaging<br>Points to improve in pouch packaging<br>Pouch product after improvement | [5,56,58] |

*3.3. Data Collection*

A survey was conducted to investigate consumer perceptions and requirements for elderly-friendly food packaging. In order to increase the response rate, we randomly selected 10 food companies and 10 nursing hospitals in Chungbuk and Daegu, South Korea and surveyed those hospitals after visiting. The questionnaire was completed by each respondent, without assistance, in the presence of the interviewer. The survey was conducted from 20 July to 20 August 2017. Table 2 shows the demographic characteristics of the respondents.

**Table 2.** Demographic characteristics.

| Characteristics | Number (n = 130) | Frequency (%) |
|---|---|---|
| **Gender** | | |
| Male | 28 | 21.5 |
| Female | 102 | 78.5 |
| **Age** | | |
| 20s | 18 | 13.9 |
| 30s | 23 | 17.7 |
| 40s | 39 | 30.0 |
| 50s | 38 | 29.2 |
| 60s | 12 | 9.2 |
| **Job** | | |
| Food company worker | 49 | 37.7 |
| Nurse | 38 | 29.2 |
| Caregiver | 43 | 33.1 |
| **Resident** | | |
| Chungbuk | 94 | 72.3 |
| Daegu | 33 | 25.4 |
| Others | 3 | 2.3 |

Notably, the end-user of the food packaging is not necessarily elderly, as the person opening or preparing a certain type of food is often a relative or other caregiver [59].

Therefore, this study also included other age groups among the respondents. However, this survey was primarily designed as a survey for the elderly, which is explicitly highlighted in the questions for each item. Nevertheless, it is sometimes recommended to approach the design of new products for elderly customers without explicitly excluding younger customers [59].

## 4. Results

### 4.1. Results of the Survey

This subsection presents the results of the survey conducted on consumers' basic perceptions of elderly-friendly food packaging. Specially, this study focuses on factors influencing the purchasing decisions of elderly-friendly food. As the first question, the survey asked the following question: Have you ever heard of elderly-friendly food? We found that 80% of respondents had never heard of such food, which was interpreted as a lack of publicity for elderly-friendly food. Table 3 presents a summary of the answers regarding the most important factor in elderly-friendly food.

**Table 3.** Answers to the question: What do you think is the most important factor in elderly-friendly food?

|  | Easy to Handle | Easy to Digest | Easy to Eat | Supply of Nutrients |
|---|---|---|---|---|
| Percentage (%) | 6 | 25 | 25 | 44 |

In total, 44% of respondents said that 'supply of nutrients' was the most important factor for elderly-friendly foods. Next, respondents chose 'easy to digest' (25%), 'easy to eat' (25%), and 'easy to handle' (6%). In general, respondents placed importance on taste, nutrition, and factors related to using the food. However, functional aspects were also important for elderly-friendly foods.

Table 4 presents the answers to the following question: 'Do you think separate food packaging materials are needed for the elderly?'. Seventy-seven percent of respondents said such materials were 'very necessary' or 'necessary'. Thus, there was a consensus in the results on the packaging of elderly-friendly foods.

**Table 4.** Answers to the question: Do you think separate food packaging materials are needed for the elderly?

|  | Very Unnecessary | Unnecessary | Normal | Necessary | Very Necessary |
|---|---|---|---|---|---|
| Percentage (%) | 2 | 5 | 16 | 50 | 27 |

The survey responses for beverage packaging preferences were as follows: pouch packaging (44%), PET packaging (25%), canned packaging (16%), and bottle packaging (15%). Thus, consumers were found to prefer packaging for the elderly that is light and easy to handle.

The respondents prefer to drink beverages in the following ways: 'using a straw' (41%), 'drinking with the mouth' (35%), and 'drinking from a cup' (24%). In the case of processed foods, the respondents prefer to eat the food through a straw or directly without using a separate container. These results demonstrated the importance of 'convenient to drink' as a factor in the NPD of beverages.

Responses to factors that should be considered first when designing packaging for elderly-friendly food were as follows: handling convenience (64%), cooking convenience (31%), and transport convenience (5%). Considering the physical limitations of the elderly, elderly-friendly foods should be convenient to transport and use.

We also evaluated consumer complaints regarding pouch packaging, which is the most common type of packaging for elderly-friendly food, and considered points to be improved. Table 5 presents the results of evaluating the inconvenience of using pouch packaging. The responses were as follows: easily overflows (73.8%), difficult to open

(47.7%), difficult to pour (47.0%), residue remains after pouring (45.4%), and difficult to use a straw (45.3%).

**Table 5.** Answers to the question: What are the disadvantages of using pouch packaging?

|  | Strongly Disagree | Disagree | Neutral | Agree | Strongly Agree |
|---|---|---|---|---|---|
| Easily overflows | 0.8 | 6.2 | 19.2 | 50.8 | 23.0 |
| Difficult to use a straw | 5.4 | 10.8 | 38.5 | 41.5 | 3.8 |
| Residue remains after pouring | 4.6 | 14.6 | 35.4 | 33.9 | 11.5 |
| Difficult to pour | 6.1 | 8.5 | 38.5 | 38.5 | 8.5 |
| Difficult to open | 9.2 | 10.0 | 33.1 | 39.2 | 8.5 |

Table 6 shows the results of the study on the aspects that should be improved in pouch packaging intended for the elderly. Improvement requests for pouch packaging were as follows: The packaging should prevent overflowing when gripping it (89.2%), the straw should be easy to use (86.9%), the contents should be easy to pour (86%), the packaging should be easy to open (80%), and residue should be minimized after pouring the product (71%). It is necessary to find a way to improve these problems simultaneously. In terms of whether improving pouch packaging by reflecting the user's requirements would increase purchase intention, 89% of the respondents answered that they would be willing to purchase the product.

**Table 6.** Answers to the following question: What needs to be improved the most in pouch-based food packaging for the elderly?

|  | Strongly Disagree | Disagree | Neutral | Agree | Strongly Agree |
|---|---|---|---|---|---|
| Preventing overflow | 0 | 0 | 10.8 | 29.2 | 60.0 |
| Using a straw easily | 0 | 0 | 13.1 | 46.1 | 40.8 |
| Minimizing residue after pouring | 0 | 3.1 | 26.2 | 39.2 | 31.5 |
| Pouring easily | 0 | 0 | 13.8 | 48.5 | 37.7 |
| Opening easily | 0.8 | 0 | 19.2 | 31.5 | 48.5 |

*4.2. Problem Solving through TRIZ*

TRIZ proceeds with the following processes: definition of the problem, selection of tools, and creation and evaluation of the solution [60]. The problems of pouch packaging for elderly people, as determined through the survey, were as follows: (i) difficult to pour, (ii) residue remains after pouring, (iii) difficult to use a straw, and (iv) easily overflows when gripped. The root of these problems is a lack of strength when the user holds the pouch container. If this problem is solved, the other problems could also be solved simultaneously. This problem occurs because the pouch packaging is difficult to grasp due to its flexibility. Thus, an imbalance between internal and external pressure occurs in a closed container. Consequently, when the packaging material is partially opened, the contents do not easily pour out well, and the probability increases that residue will remain after pouring the content. Moreover, pouch-based packaging features no guide to secure a straw, unlike pack-based packaging, and pouches cannot be grasped with force like a can or a bottle. When holding a pouch with force, the packaging container becomes deformed, and the contents overflow.

Thus, parameter 13 (stability of object composition) should be enhanced to improve pouch packaging for the elderly. However, if the stability of object's composition is strengthened, parameter 12 (shape) of the existing pouch packaging will also change, and the commercial value of the product will decrease. Thus, when one parameter is improved, the other is worsened, leading to a technical contradiction. Technical contradictions can be resolved by locating the corresponding inventive principles in the contradiction matrix [61].

In this problem, the parameter to be improved is the stability of the object composition, and the parameter that is worsened is the shape. At the point where the two parameters intersect in the contradiction matrix, the inventive principles were confirmed as follows: 1. segmentation, 22. blessing in disguise, 18. mechanical vibration, and 4. asymmetry. Among them, the most feasible principles for pouch-based food packaging were judged to be 1. segmentation and 22. blessing in disguise. Table 7 shows the TRIZ contradiction matrix for this case.

**Table 7.** Partial cells of the TRIZ Contradiction Matrix.

| IP \ WP | ... | 12. Shape | ... |
|---|---|---|---|
| ... | ... | | ... |
| 13. Stability of object composition | | 1. segmentation, 22. blessing in disguise, 18. mechanical vibration, 4. asymmetry | ... |
| ... | ... | ... | ... |

Note: IP, Improved Parameter; WP, Worsened Parameter.

Of the 40 inventive principles, principle 1, segmentation, involves dividing a fragmented system or object into independent parts, this making the system easier to disassemble. Good examples of this principle include baking cupcakes rather than large cakes and grinding hard-to-swallow pills to make them easier to consume. Principle 22, blessing in disguise refers to deriving benefits from harmful factors. Thus, this principle means 'converting harm into benefit' or 'using harmful factors to achieve a positive effect'. Examples of this principle include an allergic individual developing tolerance to an allergen through exposure to that allergen and setting a counter fire to extinguish a wildfire.

As shown in Figure 2, the packaging container was divided into parts with and without contents. Principle 1, segmentation, was applied in this process. Although it seemed unlikely that the drink would overflow, we judged that there would be a significant loss of contents and secondary problems in the process of drinking. Next, we applied principle 22, blessing in disguise. As shown in Figure 3, this principle allowed us to use the overflow phenomenon by inserting a device under the opening.

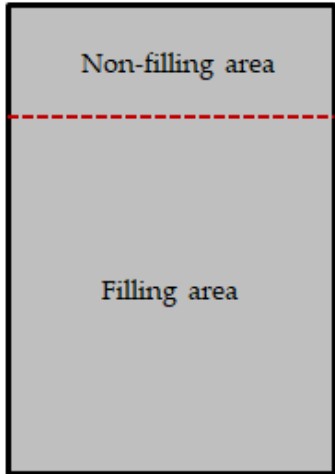

**Figure 2.** Application of principle 1, segmentation.

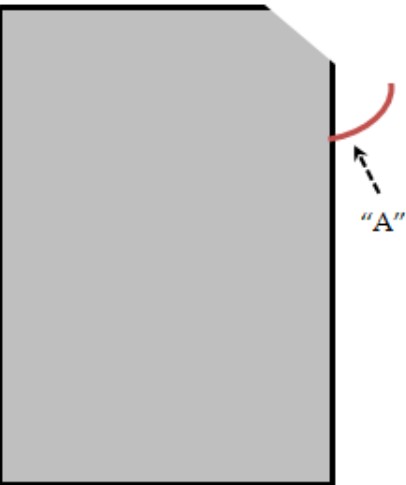

**Figure 3.** Application of principle 22, blessing in disguise ("A" is a device that receives overflowing beverages when drinking).

Portions of the upper parts of both corners lacking contents were partially laminated. Thereafter, a perforation line was added to form a finger hook when drinking, and two holes were introduced so that the contents could be consumed. Using this design, when drinking, the product is held in the hand using a finger hanger, and the contents can be drunk using a mouthpiece.

We created two drinking holes for the following reason. If only one hole is installed on one side, the contents will flow irregularly due to the pressure difference between the inside and outside of the pouch container when drinking. This type of packaging would also leave significant residue. In the end, if two holes are made on both sides, air flows through one side, reducing the pressure difference between the inside and outside. Through this action, the contents can be stably ingested with the other side. In addition, since the contents are easily discharged, there is little residue. Figure 4 shows the design reflecting the above improvements to the initial concept.

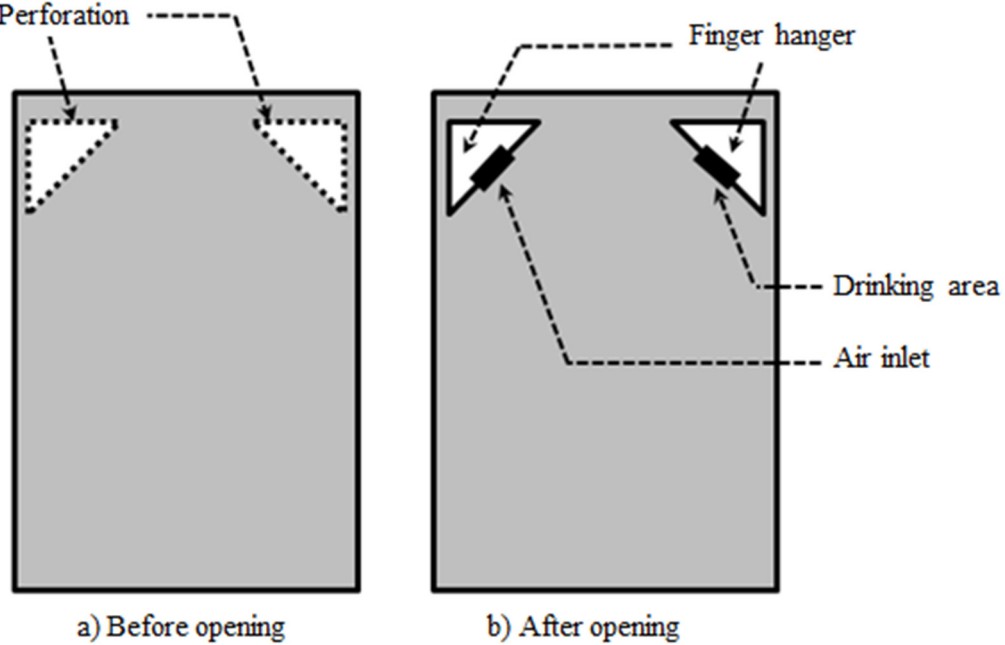

**Figure 4.** Pouch-based food package with TRIZ applied.

### 4.3. Universal Design for Elderly Friendly Food Packaging

To develop elderly-friendly food packaging, the basic principles of universal design should be considered.

First, such packaging should be developed based on the principle that elderly users with reduced physical abilities should be able to consume the food equally [Principle 1, 2]. Moreover, not only the elderly but also other physically weak people, such as the disabled and children, should be able to access and use the packaging [Principle 7]. Second, in order to use the product simply and intuitively, detailed instructions should be provided in text and figures on the food packaging [Principles 3 and 4]. Third, perforation should be applied to the opening area so that the wrapping can be opened with little physical effort [Principle 6]. Fourth, in order to reduce the risk factor of elderly users spilling, the area where the contents are discharged should be minimized [Principle 5]. Figure 5 presents a schematic diagram of applying the above universal design principles to the elderly-friendly food packaging that was initially designed on the basis of Figure 4.

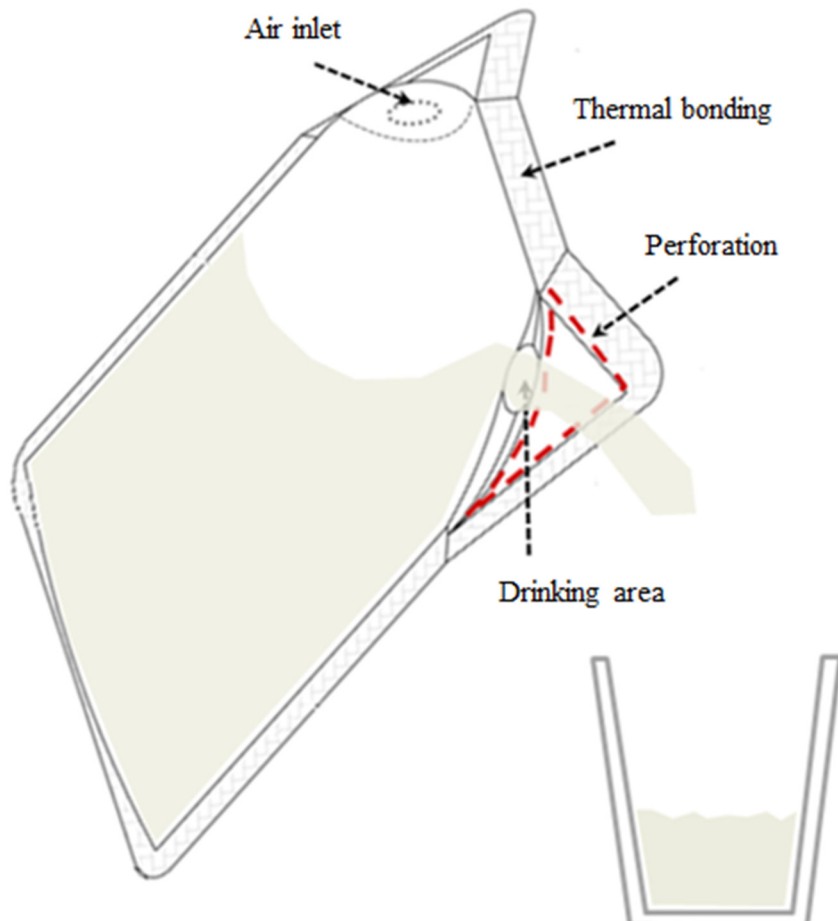

**Figure 5.** Final design of the elderly-friendly pouch-based food package.

The hatched part of the packaging is the thermal bonding part. The dotted red line at the top marks the perforation line. If the dotted line is cut in order to eat the contents, the space occupied by the dotted line is left as an empty space. This part then becomes a handle on which the user can place his or her fingers. In addition, the contents exit the container only from the cut portion in the round protruding section. This design solves the phenomenon whereby the contents overflow when holding the package. Since only the protruding round part is cut, it is easy to use a straw. Above all, when the user takes a drink, the drink comes out on one side and air enters the other side, making it possible to pour the drink without leaving any residue.

## 5. Discussion

The value of the improved design was measured by applying COMBINEX [62]. Customers value was directly allocated according to the COMBINEX method [63]. Before the process of applying the prototype, it was difficult for customers to judge the value of the product. It was hypothesized that satisfaction with these improvements would vary depending on the strength of consenting to the improvement of pouch-based food packaging for the elderly. The weight was calculated by calculating the 'strongly agree' answer ratio for each item compared to the sum of the proportions of responses to 'strongly agree' for each item in Table 6. The results in Table 5 were used to calculate the degree of satisfaction for each design. Respondents who answered 'strongly disagree', 'disagree', or 'neutral' to the question of whether they experienced disadvantages in using pouch-based packaging for each item were considered satisfied and the scores were aggregated. For the universal design, completed improvements were scored as 100, and factors not improved were scored the same as those in the old design. The results are shown in the Table 8, which outlines a COMBINEX matrix. "Criteria" are listed in the columns and "Design" is listed in the rows. In this table, 'w' refers to the weight of the value analysis, and 'Rv' indicates the related value.

**Table 8.** COMBINEX matrix to analyze the value of an improved design.

| Design \ Criteria | Preventing Overflow (w = 0.27) | | Using a Straw Easily (w = 0.19) | | Minimizing Residue after Pouring (w = 0.14) | | Pouring Easily (w = 0.17) | | Opening Easily (w = 0.22) | | Total Score |
|---|---|---|---|---|---|---|---|---|---|---|---|
| | Score | Rv | Score | Rv | Score | Rv | Score | Rv | Score | Rv | |
| Old Design | 26 | 7.14 | 55 | 10.27 | 55 | 7.93 | 53 | 9.14 | 52 | 0.22 | 46.03 |
| Universal Design | 100 | 27 | 55 | 10.27 | 100 | 14 | 100 | 17 | 52 | 0.22 | 68.49 |

Although the satisfaction level of improvement was aggregated to the maximum, the results also quantitatively confirm that the value of the product improved after applying the universal design. Ultimately, the value of the universal design product is expected to rise by 48.8%.

TRIZ is a tool that provides a practical framework [64] for open innovation and seeks to solve problems by maximizing the usage of internal and external resources in a given organization [38]. Even after acquiring various ideas, it is difficult to apply such ideas through real technology. TRIZ helps realize open innovation by providing a platform for systematic problem solving [44]. Indeed, TRIZ can be applied in all cases, regardless of the problem of closed or open innovation [65].

To gain the benefits of open innovation, TRIZ provides practical means of solving problems by focusing on the core issues [64]. To date, there have been few cases of TRIZ being applied to the development of new designs. However, implementing new technological solutions through open innovation methods such as TRIZ can afford significant cost and time savings [17,38,49], as TRIZ shortens unnecessary processes according to a certain framework.

TRIZ will be most useful for SMEs that pursue valuable innovation through new knowledge combinations and reorganization [47,48]. The TRIZ method is presently being used in the SME sector to develop fast and efficient processes, products, and sustainable services [17]. In this case, SMEs may be able to introduce innovations more rapidly. In addition, SMEs can use the TRIZ method to find ways to more effectively promote open innovation by matching the developments of innovative partners [66].

The development of elderly-friendly food packaging technology is a process for realizing social technology for the elderly, who are socially vulnerable. Moreover, this technology could serve as the basis for a sustainable business model in the social and potential market [67]. After the COVID-19 pandemic, the consumption of processed foods by the elderly will increase. Even now, the development of food packaging for the elderly is becoming an increasingly important issue. Food firms need to combine the pursuit of

commercial interests with the development of food packaging materials for the elderly, who represent a disadvantaged group [68].

To date, problem solving through the application of TRIZ has been rare in the food industry. The value chain of the food industry is complex, and various actors are working for their own interests. Since the food industry has developed for a long period of time, various problems have been exposed, but it remains difficult to solve a combination of various problems. However, it would be useful to solve complex problems through TRIZ, which can be supplemented by creating a prototype as in NPD.

## 6. Conclusions

Consumer preferences are a key to determining the success or failure of food development. This study investigated the inconveniences of pouch packaging, which is the most preferred type of food packaging. Our respondents highlighted the following issues: "The contents easily overflow when holding the container", "It is difficult to pour", "Residue remains after pouring", and "It is difficult to use a straw". We thus decided to find a method for solving all these problems at once. We then proposed a creative solution through ideas drawn from TRIZ. Ultimately, we utilized principles 1 (segmentation) and 22 (blessing and disguise) from the 40 inventive TRIZ principles. The new design for food packaging developed by universal design is not only elderly-friendly but also increases convenience in food use, regardless of age, gender, or disability [20].

This study offers the following contributions. First, the packaging industry developed from the perspective of the final producer rather than the end consumer, whereas this study approached the problem from the perspective of the consumer rather than the producer. Through a consumer survey, we identified the needs of consumers. Second, there have been few studies on elderly-friendly food packaging in the past, so, this study is expected to play a pioneering role in research on elderly-friendly food packaging. Third, our design for a new product could be directly utilized in the industry. Fourth, this study demonstrated the possibility of solving problems in the field of packaging through TRIZ. Fifth, the commercial feasibility of the new product was considered by developing the product from the perspective of universal design. In the end, our development of a new product using TRIZ highlights the importance of open innovation.

The limitations of this study are as follows. First, although this study focused on the development of elderly-friendly packaging, only nine percent of the respondents were aged 60 or older. However, it is also meaningful to collect opinions from various age groups, as this study aimed to develop a product that can be used conveniently by everyone, regardless of gender or age, through application of universal design. Second, since there is no separate standard for elderly-friendly food packaging materials, it may be difficult to label the output of this study as an elderly-friendly food packaging material. Although standards for elderly-friendly food are being prepared in various countries, there is no separate standard for elderly-friendly food packaging. Third, we used the TRIZ technique, which is uncommon in the field of packaging materials, as a method for problem solving. Specific verification will be required when using this technique to develop an actual product.

Future studies should proceed in the following direction. First, it is necessary to increase data from the elderly to determine the inconveniences experienced by the elderly when using actual food packaging and the improvements desired by that group. Second, elderly-friendly food packaging should be clearly defined. It is necessary to establish a concept that can encompass the definitions used in each country by governments and/or reputable certification bodies for aging-friendly food packaging materials. Third, the practicality of the product should be verified through prototype production. It will also be necessary to test the prototype on target customers and conduct a value evaluation using quantitative tools such as COMBINEX [60]. The process is expected to provide practical help in improving open innovation performance.

**Author Contributions:** Conceptualization, H.J., S.L. and K.S.; methodology, S.L.; software, H.J. and S.L.; validation, H.J. and S.L.; formal analysis, S.L.; investigation, H.J. and S.L.; resources, S.L.; data curation, S.L.; writing—original draft preparation, H.J. and S.L.; writing, H.J. and S.L.; visualization, H.J. and S.L.; supervision, K.S.; project administration, K.S. All authors have read and agreed to the published version of the manuscript.

**Funding:** This research received no external funding.

**Institutional Review Board Statement:** Not applicable.

**Informed Consent Statement:** Not applicable.

**Data Availability Statement:** The data used in this study are available to other authors who require access to this material.

**Conflicts of Interest:** The authors declare no conflict of interest.

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
