# Peer review of "Development of Food Packaging through TRIZ and the Possibility of Open Innovation"

_2199-8531, doi:10.3390/joitmc7040213_

Round 1

Reviewer 1 Report

The manuscript JOItmC 1334676 reports on an experimental design/strategic plan for the development of friendly food packaging for elderly people/consumers using survey data. The manuscript is a novel contribution to the field since it applies the TRIZ hypothesis. In fact, there is limited available manuscript in the literature covering this topic. It is very important to train the elderly for handling new food packaging materials. The manuscript is well written and organized and provides sufficient information regarding the survey data. My only comment is that the Conclusion section is rather too long and needs reduction. I therefore, advise the authors to reduce the length of the Conclusion section.

Author Response

Thank you very much for all your comments. We are very glad that you give us the opportunity to revise our manuscript. We have worked hard to address your excellent and detailed comments. Based on them, we have substantially improved this manuscript. We hope you like the changes we made.

Point 1.

The manuscript JOItmC 1334676 reports on an experimental design/strategic plan for the development of friendly food packaging for elderly people/consumers using survey data. The manuscript is a novel contribution to the field since it applies the TRIZ hypothesis. In fact, there is limited available manuscript in the literature covering this topic. It is very important to train the elderly for handling new food packaging materials. The manuscript is well written and organized and provides sufficient information regarding the survey data. My only comment is that the Conclusion section is rather too long and needs reduction. I therefore, advise the authors to reduce the length of the Conclusion section.

Response for Point 1.(in red)

Answer

Thank you for your kind comments. Based on your comments, we have separated the discussion and conclusion sections to shorten the conclusion section. The discussion consisted of a discussion about TRIZ in packaging design and a discussion about Triz and open innovation in design. The conclusion consisted of summary, implication, limits, and future study.

Modification #1 (line 471 – line 510 in revised manuscript)

  1. Conclusion

The success or failure of food development is a key to reflecting consumer preferences. This study investigated the inconveniences of pouch packaging, which is the most preferred food packaging. The respondents responded as follows: “It easily overflows when they hold the container”, “It is difficult to pour”, “Residue remains after pouring”, and “It is difficult to use a straw”. It was decided to find a method of solving all the problems at once. This study proposed the creative solution through the ideas drawn TRIZ. This study utilized the principles of ‘segmentation’ and ‘blessing and disguise’, of the 40 inventive TRIZ principles. New design of food packaging developed by Universal Design is not only for elderly-friendly, but also increase convenience in food use regardless of age, gender, or disability [68].

This study has the following contributions. First, packaging industry is developed from the perspective of the final producer rather than the end consumer, and this study approached the problem from the perspective of the consumer rather than the producer. Through consumer survey, this study identified the needs that consumers have. Second, there have been few studies on elderly-friendly food packaging in the past, and this study is expected to play a pioneering role in the research on elderly-friendly food packaging. Third, the design of the actual new product can be directly utilized in the industry. Fourth, it showed the possibility of solving problems in the field of packaging through TRIZ. Fifth, the commercial feasibility of new products was considered by developing new products from the perspective of universal design. In the end, new product development through TRIZ contributed to creating a new case for open innovation.

The limitations of this study are as follows. First, although this study is about the development of elderly-friendly food, only 9% of the respondents were aged 60s. However, it is also meaningful to collect opinions of various age groups because this study aims to develop a product that can be used conveniently regardless of gender and age by advocating universal design. Second, since there is no separate standard for elderly-friendly food packaging materials, it may be difficult to name the output of this study as an elderly-friendly food packaging material. Although standards are being prepared for each country for elderly-friendly food, there is no separate standard for elderly-friendly food packaging. Third, the TRIZ technique, which is not common in the field of packaging materials as method of problem solving, was used. Specific verification is required when implementing an actual product.

Future study should proceed in the following direction. First, it is necessary to increase data from the elderly. It is necessary to find out the inconveniences experienced by the elderly while using actual food packaging and the improvements they want. Second, elderly-friendly food packaging should be clearly defined. It is necessary to establish a concept that can encompass the definition of aging-friendly food packaging materials used by governments or reputable certification bodies in each country. Third, the practicality of the product is verified through prototype production.

Reviewer 2 Report

Authors describe a solution for a forbidden problem related to make more friendly the food package for elderly people. 

Comments for improving the paper:

State of the art in section 1. Introduction should include also:

  • The relation between package choice from the point of view of health.
    • In general and,
    • from elderly people point of view.
  • Creativity techniques and justification about the use of TRIZ instead other creativity techniques (brainstorming, SCAMPER,ASIT,…). As an example, there are other TRIZ derived techniques as TRIZ10:

J. D. Cano-Moreno and J. M. Cabanellas Becerra, «TRIZ10. THE DECALOGUE OF TECHNICAL CREATIVITY», DYNA, vol. 93, n.º 6, nov. 2018. Available: https://www.revistadyna.com/search/triz10-the-decalogue-of-technical-creativity

  • Why authors use Universal Design technique instead of others (design thinking, design for X,…). Compare this technique with some of them.

Lines 44-45, “However, to serve the purpose of food packaging, it is often very robust and producer-oriented”. Please, add a reference to support this claim.

Line 109-116, please add some references about TRIZ.

Line 119, please change ‘contradiction table’ by ‘contradiction matrix’.

Lines 126 to 133, should be moved after the contradiction formulation, approximately lines 253…

Lines 134 to 138 should be moved where TRIZ state of the art is described in introduction section.

Explain in Table 3, why the population includes only 9.2% of elderly people. Authors should justify properly this age distribution at this point, more detailed than the discussion of section 5.

Line 253, technical contradiction should be formulated based on the 39 parameters of the contradiction matrix, thus, readers could follow the research presented. This formulation will give the suggested principles to solve the technical contradiction.

A final comparison between the improved design and the solution would improve the understanding of the obtained advantages. I suggest authors to use COMBINEX method based on functions of the product (value analysis). The results of this comparison should be included in conclusion section too.

Author Response

Thank you very much for all your comments. We are very glad that you give us the opportunity to revise our manuscript. We have worked hard to address your excellent and detailed comments. Based on them, we have substantially improved this manuscript. We hope you like the changes we made.

Authors describe a solution for a forbidden problem related to make more friendly the food package for elderly people.

Comments for improving the paper:

Point 1.

State of the art in section 1. Introduction should include also:

The relation between package choice from the point of view of health.

In general and,

from elderly people point of view.

Response for Point 1. (in red)

Answer

Thank you for your comment. As you commented, we have added views on package choice from a general perspective, the point of view of health, and an elderly people point of view in the ‘1. Introduction’ section. This revision helped to clarify the direction of our manuscript.

Modification #1 (line 50 – line 62 in revised manuscript)

In general, packaging protects the product contained therein and provides a safe and healthy way to transport or store it [8]. In particular, the package plays a very important role in the purchase decision [9,10]. Food packaging, which is directly related to health, protects food from foreign materials, prevents spoilage, and makes eating easier. It retains the benefits of food processing after the manufacturing is complete, allowing food to travel safely from origin to point of consumption [11]. It is not otherwise separated prior to use, but rather helps its contents, food, to be absorbed into the body. Therefore, food packaging is important for maintaining food quality and safety [12]. In addition, it is necessary to consider user convenience in the development of food packaging [13]. In food packaging for the elderly, fundamentally reduced physical abilities must be taken into account. Food consumption by elderly people is mainly identified as a problem; unpacking [14] and reading labels [15]. In terms of food use, it is necessary to derive problems that occur during the actual eating process and to solve the problems.

Point 2.

Creativity techniques and justification about the use of TRIZ instead other creativity techniques (brainstorming, SCAMPER,ASIT,…). As an example, there are other TRIZ derived techniques as TRIZ10:

  1. D. Cano-Moreno and J. M. Cabanellas Becerra, «TRIZ10. THE DECALOGUE OF TECHNICAL CREATIVITY», DYNA, vol. 93, n.º 6, nov. 2018. Available: https://www.revistadyna.com/search/triz10-the-decalogue-of-technical-creativity

Response for Point 2. (in red)

Answer

Thank you for this comment. According to your comment, we considered various creativity techniques such as brainstorming, SCAMPER, ASIT. And we presented arguments for we chose TRIZ. This modification justified the use of the TRIZ method in this study.

Modification #2 (line 150 – line 158 in revised manuscript)

There are various methods of problem-solving techniques; Brainstorming, SCAMPER, ASIT and etc. Brainstorming popularized by Osborn [30] is one of the meeting techniques in which group members try to find solutions to problems by freely presenting ideas. It turns out that brainstorming groups have fewer ideas and are of poorer quality than the same number of individuals working alone [31]. This technique has difficulties in generating innovative ideas step by step [32]. Advanced Systematic Inventive Thinking (ASIT) is a problem solving technique derived from TRIZ and developed to be easily utilized in the field [33]. Because solutions are sought using objects available in the problem space of the system [34], it could be difficult to pursue open innovation.

Point 3.

Why authors use Universal Design technique instead of others (design thinking, design for X,…). Compare this technique with some of them.

Response for Point 3. (in red)

Answer

Thank you for this kind comment. In previous version, we missed the justification of using Universal Design technique. According to your comment, we reviewed other design techniques. There are design thinking and design for X. We investigated the characteristics of the techniques. We found the advantages of only Universal design. This revision help to strengthen the structure of this study.

Modification #3 (line 128 – line 148 in revised manuscript)

Theories for design improvement include ‘design thinking’ and ‘design for X’(DFX). Design thinking is generally defined as an analytical and creative process that engages people in experimentation, model creation and prototyping, feedback, and redesign opportunities [23]. The main idea is that the way professional designers solve problems is valuable to innovating companies and changing societies [24]. However, according to Kimbell [24], design thinking has problems; First, it relies on a dualism among thinking, knowing, and acting in the world. Also he argued generalized design thinking ignores the historically established diversity of designer practices and institutions. Third, design thinking relies on design theory that privileges designers as major actors in design. The DFX is a generic framework that can be easily extended or tuned for rapid development of various DFX tools with consistent quality, and provides many formal but practical structures [25]. DFX techniques are widely used in holistic product development, but each focuses only on one aspect of the product such as manufacturing or cost [26]. Universal design is the design of products to be usable by as many people as possible at little or no extra cost [27]. It indicates that the use of universal design principles along with the open innovation concept can increase the possible sustainability goals of information systems, and this can be done by improving the success of the system along with increased user satisfaction [28]. Since the customer base becomes the disabled, the elderly, and the socially disadvantaged, companies can maximize the potential market by using universal design from the early stage of product development [29]. Therefore, this study judged that universal design is suitable for businesses targeting the elderly.

Point 4.

Lines 44-45, “However, to serve the purpose of food packaging, it is often very robust and producer-oriented”. Please, add a reference to support this claim.

Response for Point 4. (in red)

Answer

Thank you for your comment. According to your comment, we add a reference for this argument. The reference is a study of Kirwan and Strawbridge (2003). They proposed the purpose of food packaging for the robustness and producer oriented characteristic.

Modification #4 (line 46 – line 47 in revised manuscript)

However, to serve the purpose of food packaging, it is often very robust and producer-oriented [7].

Modification #5 (line 534 in revised manuscript)

[7] Kirwan, M.J.; Strawbridge, J.W. Plastics in Food Packaging. Food packaging technology 2003, 1, 174-240.

Point 5.

Line 109-116, please add some references about TRIZ.

Response for Point 5. (in red)

Answer

Thank you for your comment. As you commented, we add references about TRIZ in section 2.3. Basically, Altshuller’s study, the inventor of TRIZ, was added. And we added Su, Lin, and Chiang(2008).

Modification #6 (line 159 – line 160 in revised manuscript)

TRIZ (Teoriya Reshniya Izobretatelskikh Zadatch, TRIZ) is a Russian abbreviation for the theory of inventive problem solving [35].

Modification #7 (line 165 – line 167 in revised manuscript)

TRIZ promotes the thinking of researchers to derive creative solutions, and is very useful in solving scientific and technological problems because it is a problem-solving methodology created based on invention patents [36].

Modification #8 (line 165 – line 167 in revised manuscript)

The existence of principles means that we have a theoretical framework that can specifically explain the problem and help engineers find creative idea [35].

Modification #9 (line 581 in revised manuscript)

Altshuller, G. 40 Principles: TRIZ Keys to Innovation.; Technical Innovation Center, Inc., 2002.

Modification #10 (line 582 – line 583 in revised manuscript)

Su, C.; Lin, C.; Chiang, T. Systematic Improvement in Service Quality through TRIZ Methodology: An Exploratory Study. Total Qual. Manage. 2008, 19, 223-243.

Point 6.

Line 119, please change ‘contradiction table’ by ‘contradiction matrix’.

Response for Point 6. (in red)

Answer

Thank you for your comment. According to your comment, we changed it by ‘contradiction matrix’

Modification #11 (line 168 – line 170 in revised manuscript)

TRIZ studied important patents around the world, set 39 engineering parameters having a technical contradiction relationship, and created a ‘contradiction matrix’ that lists these parameters horizontally and vertically [37].

Point 7.

Lines 126 to 133, should be moved after the contradiction formulation, approximately lines 253…

Response for Point 7. (in red)

Answer

Thank you for your comment. According to your comment, we moved the part to the point after contradiction formulation.

Modification #12 (line 338 – line 345 in revised manuscript)

Of the 40 inventive principles, the principle of ‘segmentation’ divides a fragmentation system or object into independent parts, making it easier to disassemble. A good example is making cupcakes rather than large cakes, or grinding hard-to-swallow pills to make them easier to take. The principle of ‘blessing in disguise’ is the principle of taking advantage from harmful factors. It means ‘converting harm into benefit’ or ‘using harmful factors to achieve a positive effect’. A good example is exposure to an extract of the same substance to develop tolerance to an allergen that causes an allergy, or setting a counter fire to put out a wildfire.

Point 8.

Lines 134 to 138 should be moved where TRIZ state of the art is described in introduction section.

Response for Point 8. (in red)

Answer

Thank you for your comment. As you commented, we moved the part to the point where TRIZ state of the art in introduction section. And we we modified the first part of the following to make the flow of the sentence natural.

Modification #13 (line 66 – line 71 in revised manuscript)

Wan et al. [16] applied the TRIZ theory to a new energy vehicle service and formed an idea according to service derivative characteristics. Their research validates the solution by examining the practical problems in developing the new energy vehicle market. Feniser et al. [17] used TRIZ to obtain efficient processes, products and sustainable processes for small and medium sized enterprise (SMEs).

Modification #14 (line 71 – line 74 in revised manuscript)

In the context of TRIZ research, this study will contribute to further catalyzing the development of the elderly-friendly food industry while making food consumption more convenient for the elderly by developing a pouch packaging design for the elderly.

Point 9.

Explain in Table 3, why the population includes only 9.2% of elderly people. Authors should justify properly this age distribution at this point, more detailed than the discussion of section 5.

Response for Point 9. (in red)

Answer

Thank you for your comment. What you commented in this point was perhaps the weakest part in this study. Authors thought that the elderly should be surveyed for this study in the first time. However, the users of the food package do not necessarily have to be elderly users. The idea made it possible to expand the age spectrum of the targets. It should be noted that the elderly may be able to get help from relatives or caregivers when it comes to eating. A person who helps the elderly in food intake or observes it can rather objectively talk about the problem of food packaging. Since this study is oriented toward a study based on universal design, it has helped justify the investigating regardless of age. According to your comment, we added the statement for justify the age distribution in survey with fewer elderly people.

Modification #15 (line 269 – line 274 in revised manuscript)

The user of a food package is not necessarily an elderly end-user and the person operating a certain food may often be a relative or other caregiver [59]. Therefore, this study included other age groups in the respondents. It is premised that this survey is a survey for the elderly, and it is explicitly revealed in the questions of each item. It is sometimes recommended to approach the design of new product development for elderly customers without explicitly excluding younger customers [59].

Point 10.

Line 253, technical contradiction should be formulated based on the 39 parameters of the contradiction matrix, thus, readers could follow the research presented. This formulation will give the suggested principles to solve the technical contradiction.

Response for Point 10. (in red)

Answer

We sincerely thank you for this comment. We also hope that readers will be helped in resolving technical contradictions as this study progresses.

Point 11.

A final comparison between the improved design and the solution would improve the understanding of the obtained advantages. I suggest authors to use COMBINEX method based on functions of the product (value analysis). The results of this comparison should be included in conclusion section too.

Response for Point 11. (in red)

Answer

Thank you for this comment. As you proposed, we applied Combinex method to the improved design and previous version. The results helped us to quantify the value of the improved product and old version. However, the improved value should be evaluated by the customer. In this study, it was calculated by assumptions and calculations. Therefore, in future research, we plan to conduct a prototype-based valuation study using COMBINEX.

Thank you once more for all your valuable contributions to our paper. Your detailed recommendation improves this manuscript.

Modification #16 (line 413 – line 432 in revised manuscript)

The value of the improved design was measured by applying COMBINEX [60]. According to the COMBINEX method, the customers directly allocate the value [61]. Before the process of applying the prototype, it was difficult for customers to judge the value. It was hypothesized that satisfaction with the improvement will vary depending on the strength of consent to the improvement of pouch food packaging for the elderly. The ‘weigh’ was calculated by calculating the ‘strongly agree’ answer ratio for each item compared to the sum of the proportions of responses to ‘strongly agree’ to each item in Table 7. The results in Table 6 were used to calculate the degree of satisfaction for each design. Respondents who answered ‘strongly disagree’, ‘disagree’, ‘neutral’ to the question of whether they experienced disadvantage in using pouch packaging for each item were considered satisfied and the scores were aggregated. For the universal design, improvements completed were scored as 100, and those not improved were scored the same as the old design. The results are shown in the table below. Table proposes a COMBINEX matrix. “Ctiteria” are listed in the columns and “design” is listed in the rows. In the table, ‘w’ refer to weigh of value analysis, ‘Rv’ indicates the related value.

Table 8. COMBINEX matrix for analysis of value for improved design

Criteria

Design

Preventing overflow

(w=0.27)

Using a straw easily

(w=0.19)

Minimizing residue after pouring

(w=0.14)

Pouring easily

(w=0.17)

Opening easily

(w=0.22)

Total Score

Score

Rv

Score

Rv

Score

Rv

Score

Rv

Score

Rv

Old

Design

26

7.14

55

10.27

55

7.93

53

9.14

52

0.22

46.03

Universal

Design

100

27

55

10.27

100

14

100

17

52

0.22

68.49

Although the satisfaction level of improvement was aggregated to the maximum, it can be quantitatively confirmed that the value of the product has improved after applying the universal design through the results. The value of the universal design product is expected to rise by 48.8%.

Modification #17 (line 510 – line 513 in revised manuscript)

It is necessary to test the prototype on target customers and conduct a value evaluation using quantitative tools such as COMBINEX [60]. The process is expected to give practical help in creating open innovation performances.

Reviewer 3 Report

Congratulations to the authors of an interesting article. Undoubtedly, the problem described in the article and the concepts of its solution can serve as material for further research.
The subject of the article indicates that the research should be conducted mainly among the elderly. Seniors are most competent to answer. In the authors' research, only 38.4% of the respondents were over 50 years old. This issue was taken into account in the limitations of the study, which indicates that the authors are very aware of the shortcomings of the study.
The required correction of the table number should be: Table 2 (line 178).
The Reviewer's recommendation is to continue the research among the elderly and to use the packaging prototype in the research.
The research should take into account various types of seniors' dysfunctions, their individual characteristics and the conditions in which the products are consumed.

Author Response

Thank you very much for all your comments. We are very glad that you give us the opportunity to revise our manuscript. We have worked hard to address your excellent and detailed comments. Based on them, we have substantially improved this manuscript. We hope you like the changes we made.

Congratulations to the authors of an interesting article. Undoubtedly, the problem described in the article and the concepts of its solution can serve as material for further research.

Point 1.

The subject of the article indicates that the research should be conducted mainly among the elderly. Seniors are most competent to answer. In the authors' research, only 38.4% of the respondents were over 50 years old. This issue was taken into account in the limitations of the study, which indicates that the authors are very aware of the shortcomings of the study.

Response for Point 1. (in red)

Answer

Thank you for your comment. We are aware of the limitation of the age distribution of the survey targets in this study. This study tried to justify properly this age distribution.

Modification #1 (line 269 – line 274 in revised manuscript)

The user of a food package is not necessarily an elderly end-user and the person operating a certain food may often be a relative or other caregiver [59]. Therefore, this study included other age groups in the respondents. It is premised that this survey is a survey for the elderly, and it is explicitly revealed in the questions of each item. It is sometimes recommended to approach the design of new product development for elderly customers without explicitly excluding younger customers [59].

Point 2.

The required correction of the table number should be: Table 2 (line 178).

Response for Point 2. (in red)

Answer

Thank you for your comment. We revised it correctly.

Modification #2 (line 257 – line 258 in revised manuscript)

Table 2 summarizes the measurement items based on the relevant literature for each factor.

Point 3.

The Reviewer's recommendation is to continue the research among the elderly and to use the packaging prototype in the research.

Response for Point 3. (in red)

Answer

Thank you for your comment. Based on your comment, this study proposed the necessity of future study using prototype.

Modification #3 (line 510 – line 513 in revised manuscript)

It is necessary to test the prototype on target customers and conduct a value evaluation using quantitative tools such as COMBINEX [60]. The process is expected to give practical help in creating open innovation performances.

Point 4.

The research should take into account various types of seniors' dysfunctions, their individual characteristics and the conditions in which the products are consumed.

Response for Point 4. (in red)

Answer

According to your comment, this study additionally stated that it was necessary to expose and solve the problem between the decrease in physical ability and food use from the perspective of the elderly in the ‘1.Itroduction’ section.

Modification #4 (line 59 – line 62 in revised manuscript)

In food packaging for the elderly, fundamentally reduced physical abilities must be taken into account. Food consumption by elderly people is mainly identified as a problem; unpacking [14] and reading labels [15]. In terms of food use, it is necessary to derive problems that occur during the actual eating process and to solve the problems.

Round 2

Reviewer 2 Report

Authors have addressed in a right way the comments except one, Point 10, a key point.
 It is very important for readers that the authors formulate explicitly the technical contradiction based on the 39 parameters, If parameter X is improved, the parameter Y gets worse, this formulation allows to go into the contradiction matrix, and here readers should find the suggested inventive principles to use (it suppossed segmentation, blessing and disguise should be among them).

Author Response

Thank you very much for all your comments. We are very glad that you give us the opportunity to revise our manuscript again. We have worked hard to address your excellent and detailed comments. Based on them, we have substantially improved this manuscript. We hope you like the changes we made.

Point 1.

Authors have addressed in a right way the comments except one, Point 10, a key point.
 It is very important for readers that the authors formulate explicitly the technical contradiction based on the 39 parameters, If parameter X is improved, the parameter Y gets worse, this formulation allows to go into the contradiction matrix, and here readers should find the suggested inventive principles to use (it suppossed segmentation, blessing and disguise should be among them).

Response for Point 1.(in red)

Answer

Thank you for your kind comment. At previous revision, we misunderstood your comment as positive. In this revision, this study proposed how we used the contradiction matrix according to your comment. That process was missing from the first manuscript. On the revision, we identified improving parameter and worsening parameters by improving. This study found the inventive principles in the contradiction matrix. In the end, those help invent improved design. We have adequately described the omissions in the first manuscript based on your comments. Your comments have enhanced our research. Thank you again.

Modification #1 (line 326 – line 329 in revised manuscript)

TRIZ proceeds with the following processes; definition of the problem, selection of tools, creation and evaluation of solution [60]. The probles of pouch packaging for elderly people investigated through the survey are as follows

Modification #2 (line 341 – line 354 in revised manuscript)

‘Stability of object composition’ should be enhanced to improve pouch packaging for the elderly. However, if ‘stability of object composition’ is strengthened, the ‘shape’ of the existing pouch packaging will change and the commercial value of the product will be lowered. When one parameter is improved, the other is worsened. A technical contradiction aroused. Technical contradictions could be resolved by finding the corresponding inventive principles in the contradiction matrix [61]. In this problem, the parameter to be improved is stability of object composition, and the parameter to be worsened is shape. At the point where two parameters intersect in the contradiction matrix, the inventive principles can be confirmed as follows; ‘Segmentation’, ‘Blessing in disguise’, ‘Mechanical vibration’, and ‘Asymmetry’. Among them, the feasible principle applied to pouch food packaging was judged to be ‘Segmentation’ and ‘Blessing in disguise’.

Table 8. Partial cells of TRIZ Contradiction Matrix

IP              WP

Shape

Stability of object composition

Segmentation, Blessing in disguise, Mechanical vibration, Asymmetry

Note: IP, Improving Parameter; WP, Worsening Parameter.

Modification #3 (line 363 – line 368 in revised manuscript)

Although pouch packaging is preferred because it is light, there is a technical contradiction that causes the above problems owing to such ductility. To solve this technical contradiction, TRIZ principles were applied. In order to solve the first problem, ‘segmentation’ principle was applied. As shown in Figure 2, the packaging container was divided into parts with and without contents. Segmentation principle was applied in this process.

Round 3

Reviewer 2 Report

Authors have been well adressed the coment.

Please, include the numbers of parameters and principles.

Author Response

Thank you very much for all your comments. We are very glad that you give us the opportunity to revise our manuscript again. We have worked hard to address your excellent and detailed comments. Based on them, we have substantially improved this manuscript. We hope you like the changes we made.

Point 1.

Authors have been well addressed the comment.

Please, include the numbers of parameters and principles.

Response for Point 1.(in red)

Answer

Thank you for your kind comment. This study included the numbers of engineering parameter and inventive principle applied in the solution. Your comments have enhanced our research. Thank you again.

Modification #1 (line 10 – line 11 in revised manuscript)

To solve all these problems at once, this study applied the following principles of TRIZ '1. segmentation' and '22. blessing and disguise'.

Modification #2 (line 342 – line 354 in revised manuscript)

The parameter 13 ‘Stability of object composition’ should be enhanced to improve pouch packaging for the elderly. However, if the parameter 13 ‘stability of object composition’ is strengthened, the parameter 12 ‘shape’ of the existing pouch packaging will change and the commercial value of the product will be lowered. When one parameter is improved, the other is worsened. A technical contradiction aroused. Technical contradictions could be resolved by finding the corresponding inventive principles in the contradiction matrix [61]. In this problem, the parameter to be improved is stability of object composition, and the parameter to be worsened is shape. At the point where two parameters intersect in the contradiction matrix, the inventive principles can be confirmed as follows; ‘1. Segmentation’, ’22. Blessing in disguise’, ’18. Mechanical vibration’, and ‘4. Asymmetry’. Among them, the feasible principle applied to pouch food packaging was judged to be ‘1. Segmentation’ and ’22. Blessing in disguise’. Table 8 shows the TRIZ contradiction matrix in this case.

Modification #3 (line 355 – line 356 in revised manuscript)

Table 8. Partial cells of TRIZ Contradiction Matrix

IP              WP

12. Shape

13. Stability of object composition

1. Segmentation, 22. Blessing in disguise, 18. Mechanical vibration, 4. Asymmetry

Note: IP, Improving Parameter; WP, Worsening Parameter.

Modification #4 (line 357 – line 374 in revised manuscript)

Of the 40 inventive principles, the principle ‘1. segmentation’ divides a fragmentation system or object into independent parts, making it easier to disassemble. A good example is making cupcakes rather than large cakes, or grinding hard-to-swallow pills to make them easier to take. The principle ’22. blessing in disguise’ is the principle of taking advantage from harmful factors. It means ‘converting harm into benefit’ or ‘using harmful factors to achieve a positive effect’. A good example is exposure to an extract of the same substance to develop tolerance to an allergen that causes an allergy, or setting a counter fire to put out a wildfire.

             As shown in Figure 2, the packaging container was divided into parts with and without contents. The principle ‘1. segmentation’ was applied in this process. Although it seems unlikely that the drink will overflow, it was judged that there was a lot of loss of contents and secondary problems occurred in drinking. Next, the principle’22. blessing in disguise’ is applied. As shown in Figure 3, it made it possible to use the overflow phenomenon by inserting a device under the opening.

Modification #5 (line 375 – line 379 in revised manuscript)

Figure 2. Application of the principle ‘1. segmentation’

Figure 3. Application of the principle ’22. blessing in disguise’ (“A” is a device that receives overflowing beverages when drinking)

Modification #6 (line 498 – line 501 in revised manuscript)

This study utilized the principles  ‘1. segmentation’ and ’22. blessing and disguise’ of the 40 inventive TRIZ principles. New design of food packaging developed by Universal Design is not only for elderly-friendly, but also increase convenience in food use regardless of age, gender, or disability [70].
